# Design and Validation of a Variable-Rate Control Metering Mechanism and Smart Monitoring System for a High-Precision Sugarcane Transplanter

Abdallah E. Elwakeel [1,*], Yasser S. A. Mazrou [2], Ahmed S. Eissa [3], Abdelaziz M. Okasha [4], Adel H. Elmetwalli [5], Abeer H. Makhlouf [6], Khaled A. Metwally [7], Wael A. Mahmoud [8] and Salah Elsayed [9,10]

1 Agricultural Engineering Department, Faculty of Agriculture and Natural Resources, Aswan University, Aswan 81528, Egypt
2 Applied College, Muhayil Asir, King Khalid University, Abha 62587, Saudi Arabia; ymazrou@kku.edu.sa
3 Agricultural Products Process Engineering Department, Faculty of Agricultural Engineering, Al-Azhar University, Cairo 11751, Egypt; ahmedeissa2205.el@azhar.edu.eg
4 Department of Agricultural Engineering, Faculty of Agriculture, Kafrelsheikh University, Kafr El-Sheikh 33516, Egypt; abdelaziz.okasha@agr.kfs.edu.eg
5 Agricultural Engineering Department, Faculty of Agriculture, Tanta University, Tanta 31527, Egypt; adel.elmetwali@agr.tanta.edu.eg
6 Department of Agricultural Botany, Faculty of Agriculture, Minufiya University, Shibin El-Kom 32511, Egypt; abeer.makhlof@agr.menofia.edu.eg
7 Soil and Water Sciences Department, Faculty of Technology and Development, Zagazig University, Zagazig 44511, Egypt; khametwally@zu.edu.eg
8 Department of Agricultural Machinery and Power Engineering, Faculty of Agricultural Engineering, Al-Azhar University, Assiut 71524, Egypt; waelshaban.50@azhar.edu.eg
9 Agricultural Engineering, Evaluation of Natural Resources Department, Environmental Studies and Research Institute, University of Sadat City, Sadat City 32897, Egypt; salah.emam@esri.usc.edu.eg
10 New Era and Development in Civil Engineering Research Group, Scientific Research Center, Al-Ayen University, Nasiriyah 64001, Thi-Qar, Iraq
* Correspondence: abdallah_elshawadfy@agr.aswu.edu.eg

**Abstract:** The current study aimed to design and test the accuracy of a variable-rate control metering mechanism (VRCMM) and a remote smart monitoring system (RSMS) for a precision sugarcane transplanter based on IoT technology. The VRCMM is used to operate the seedling metering device at different speeds using a stepper motor based on the travel speed, and the RSMS was employed to evaluate of the three basic parameters of seedling amount, optimum rate, and missed rate. Two types of sensors were used for detecting the sugarcane seedling (SS) and travel speed, including one ultrasonic sensor and one infrared RPM sensor. The study was performed at five travel speeds and four transplant spacings. The findings of laboratory tests showed that the mean record of the relative error between the desired stepper motor speed of the VRCMM and the real value was 3.39%, and it increased with increasing the travel speed. as Additionally, the speed regulation performance was in agreement with the transplanting index. The change in RSMS accuracy is obvious when the travel speed is high and the transplant spacing is small. The RSMS accuracy drops sharply, revealing a leaping change. In conclusion, the smart and intelligent designed sugarcane transplanter would be very useful in sugarcane production.

**Keywords:** planter; seeder; metering device; ultrasonic sensor; Arduino; precision agriculture

## 1. Introduction

Worldwide, sugar is regarded as one of the most important strategic commodities, and in Egypt, it is ranked the second most important strategic commodity after wheat. Thus, it was essential to give our whole attention to sugarcane for enhancing crop yield and to

decrease the gap between consumption and crop production. In Egypt, sugarcane is grown on over 126,000 ha of agricultural land having an average production of 114.3 ton/ha, and total annual production may reach 16 million tons [1–5].

Planting technique represents one of the most crucial practices in the production of sugarcane. It results in a quickly and consistently emerging plant stand at the desired plant density. However, it takes a lot of time and work [6–8]. Mechanized field planting has become more popular in the sugarcane industry as a solution to labor shortages and expensive production [9]. Rípoli et al. [10] compared five different sugarcane planters' cost effectiveness to semi-mechanized equipment and stated that the semi-mechanized equipment was much more expensive. In addition, automatic sugarcane planting machine costs less to operate. It greatly saves labor in the working process, and operates extended hours during both day and night shifts [11]. At this time, the three primary types of sugarcane planter are whole-stalk planters, real-time planters, and billet planters [6]. In addition to other industrialized countries, the United States, Japan, and Italy, among others, have begun to investigate sugarcane transplanting and have created automatic transplanter machinery suitable for diverse geographical environments and crop varieties [12–15].

Precision agriculture (PA) is a methodical strategy for restructuring the production agricultural system in order to achieve a low-input, highly effective, and long-lasting system [16,17]. Developing a decision-support system for farm management with the goal of maximizing input return while preserving resources is PA's primary goal [18]. With the benefits of conserving seeds, lowering labor intensity, and optimizing operational efficiency, precision seeding and planting is going to be an essential direction for advancement in agriculture in the coming years [19,20]. PA is also a very important aspect for transplanting sugarcane at the optimum time, with the optimum moisture content, at the same intervals. Total grain seeds must be uniformly distributed throughout the field in addition to being planted or drilled at the ideal planting or seeding rate per area in order to produce the desired crop rows. Additionally, maintaining the ideal plant-to-plant spacing boosts crop yield [21–26]. Plant spacing and crop yield have a well-established relationship [27–29]. For numerous crops, row plant spacing is influenced just as significantly by the crop's specific biological spacing demands as it is by the structural design of the agricultural machines utilized [30]. In addition, a key point of the precise planting of a precision transplanting or seeding machine in a typical square grid pattern needs equipment calibration tests before planting in both directions of the crop row [31]. With the advancement of precision planting in smart agriculture, variable rate application (VRA), which aims to increase savings of fertilizers, seeds, and herbicides in keeping with site specific management particular administration basics within a field, has grown in popularity [32].

By modifying seeding rates based on field conditions, this crucial PA component has substantially raised agricultural production and preserved seeds while also raising yield and net farm returns [33,34].

The pace of mechanization in sugarcane planting is increasing daily. Over several decades, many researchers worldwide have created sugarcane billet measuring devices. A lot of research has been conducted regarding the uniformity of sugarcane billet planting machines. Razavi et al. [35] investigated how a sugarcane billet planter's (SBP) forward speed and the chain conveyor angle affected planting uniformity. Taghinezhad et al. [36] studied the impact of rotation angle and forward speed of SBPs on precision indexes and discharging. Saengprachatanarug et al. [37] improved the discharge mechanism of the SBP by adjusting the slope of the billet's hopper. Saengprachatanarug et al. [38] modified a metering mechanism for SBP by developing the arrangement of the cleat's conveyor. Moslem et al. [39] developed a metering mechanism capable of planting sugarcane billets at 50 cm spacing.

The foundation of the automation and intelligence of seeding and transplanting, as well as the trustworthy assurance of signal feedback and equipment research along with the development of the follow-up seeding links, has enhanced the efficiency of seeding and transplanting. Photoelectric monitoring, high-speed photography, and piezoelectric

sensing are just examples of monitoring techniques that many researchers have utilized to monitor seeding quality [26,40–43]. The most common criteria for evaluating seeding quality include seeding quantity, missed rate, and optimum rate. It is usually necessary to set up a monitoring device of seedlings in order to obtain these parameters. Furthermore, monitoring sensors are an extremely essential part of the monitoring device.

Currently, companies have created and designed different monitoring technologies such as laser diodes, optical sensors, photoresistors, and infrared sensors, as well as optical fiber sensors being examples of these technologies [21,44–46]. In addition, many researchers have employed image processing, piezoelectric sensors, and capacitive sensors to track the flow data of seeds moving via the seed tube [22,47,48]. Yet, there are no automated seedling sensors available for sugarcane or vegetable transplanters. To the best of the authors' knowledge, there are not yet many SS meters that can transplant varying amounts of sugarcane. In this regard, too much research is required to enhance sugarcane transplanting efficiency. Based on the abovementioned discussion, the current study provides technical and practical references for redesigning sugarcane precision transplanting–metering machines and also offers crucial machine innovation for the progress of precision agriculture.

The objectives of the study were to:

1. Design and assess the performance of a variable-rate control metering mechanism (VRCMM) for sugarcane transplanting based on IoT technology.
2. Design and test a remote smart monitoring system (RSMS) for a precise sugarcane transplanter based on an ultrasonic sensor, IoT technology, Android, and wireless communication.

## 2. Materials and Methods

### 2.1. SSs

A twenty-month-old C-9 sugarcane variety was used for this investigation. Applying a sugarcane bud chipping machine generated by Elwakeel et al. [5], sugarcane buds were cut off 1.5 m from the roots of healthy sugarcane plants that were free of insects and diseases. In a plastic greenhouse, the extracted sugarcane buds were planted in a plastic tray filled with farmyard manure (FYM), which is prepared basically using cow dung, cow urine, rice straw, and other crop residues. The FYM was mixed with soil at a ratio of 1:1 and grown. After 70 days of planting, the seedlings were at their ideal height of 35 cm for transplanting.

### 2.2. Overall Structure and Principles of Working

The overall structure of a variable-rate control metering mechanism (VRCMM) for the sugarcane transplanter is shown in Figure 1, and was composed of a seedling cylinder, a stepper motor, its driver, a pinion, a sprocket, a speed sensor, an Arduino mega board (AMB), and a seedling counter sensor (ultrasonic sensor). As the seedling cylinder rotated clockwise, the SSs in the rotating seedling cylinders would fill the seedling-guiding slot (SGS), and then be discharged on the earlier open ridge. To track the number of discharged seedlings, a remote smart monitoring system (RSMS) was installed at the outlet of the SGS.

As seen in Figure 1, the sugarcane transplanter is fundamentally composed of a two-wheel walking tractor and a transplanting unit. The Bluetooth unit and control unit are fixed on the upper point of the two-wheel walking tractor to receive and transmit signals for precise management, operation control, and remote monitoring. The transplanting unit is attached to the tractor via a four-point attaching and primarily consists of a furrow opener, a rotary seedling cylinder, two soil-compaction wheels, an RSMS, and a VRCMM.

### 2.3. Design of Electronic Circuits for VRCMM and RSMS

The VRCMM and the RSMS consist of six main parts: an infrared light speed sensor (model: LM 393 IC, China), an ultrasonic sensor (model: HC-SR04, Hammond Manufacturing L., Canada), an Arduino Mega (model: 2500 R3, China), a Bluetooth unit (model: HC-05, Yantel Corporation, China), stepper motors (model: Nema 23, CUI Devices, Tualatin, USA), and a stepper motor driver (model: TB 6600, Sorotec GmbH, Germany). In addition, some

axillary components, such as a laptop (model: TOSHIBA Intel (R) Core (TM) i3-3120M CPU @ 2.50 GHz, Tokyo) or smart phone (OPPO A15, OPPO, Guangdong, China), bread boards, wires, a USB cable, a PV panel (60 W, Eagle Network Supply Private Limited, India), a battery charger (12 V, Victron Energy, The Netherlands), a converter (12 V–5 V) {Optional}, and battery (70 A·h), were also used. These components are depicted and described in Figures 2–4. Figure 2 illustrates signal flowrate, electric energy connections, and mechanical connections between the different electronic parts. A schematic diagram of the VRCMM and the RSMS with successful electrical connections is shown in Figure 3. Figure 4 demonstrates the technical specifications of the main electronic components of the VRCMM and the RSMS.

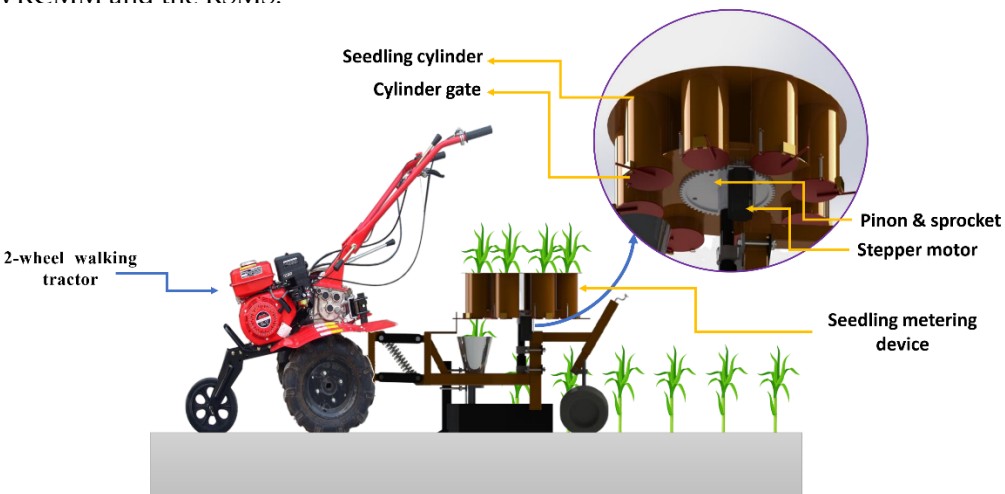

**Figure 1.** The overall structure of the VRCMM.

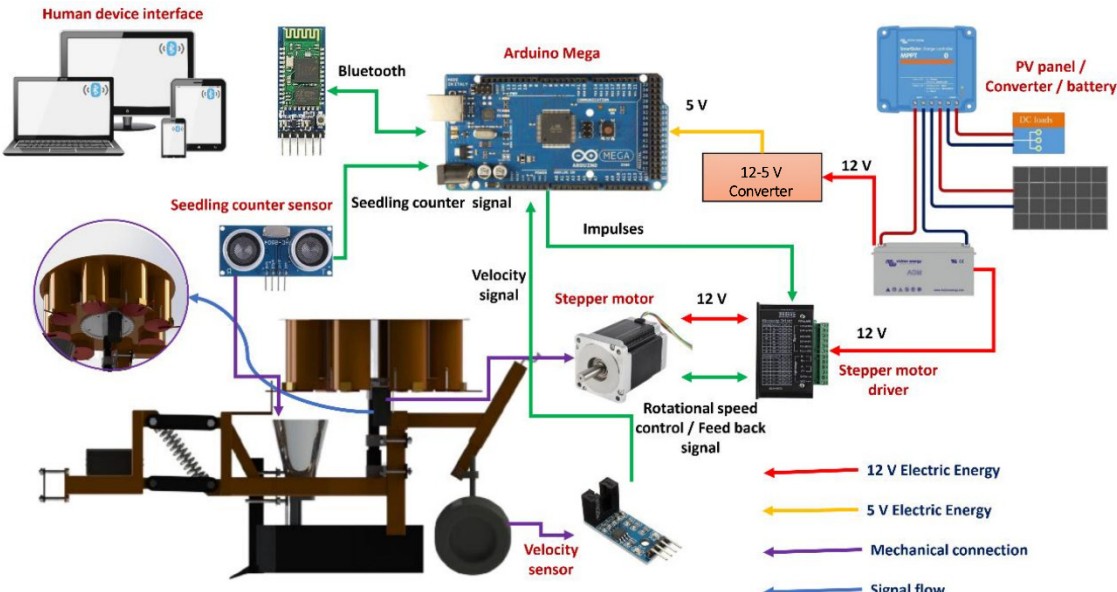

**Figure 2.** The operation map of the VRCMM and the RSMS.

### 2.3.1. Speed Sensor

The speed sensor (mode: LM393 IC) was employed to determine the ground wheel rotational speed. The speed sensor comprises of a phototransistor and an infrared LED which are fixed on a small, printed circuit board. When the LED is switched on, it releases a beam of infrared rays (IR). The phototransistor will detect any interruption of the IR beam caused by the rotating ground wheel, and produce an electrical pulse. The number of pulses per second is closely linked to the RPM of the ground wheel. In addition, using a microcontroller, it may be utilized to determine the number of pulses per second and then

convert that count to RPM, and finally to forward speed (linear speed). The speed sensor can work in a voltage range of 3.3 to 5 V. The output signal is high, and it is low when the sensor detects pulses. Figure 5 illustrates the main parts and connecting pins of the speed sensor used in the VRCMM.

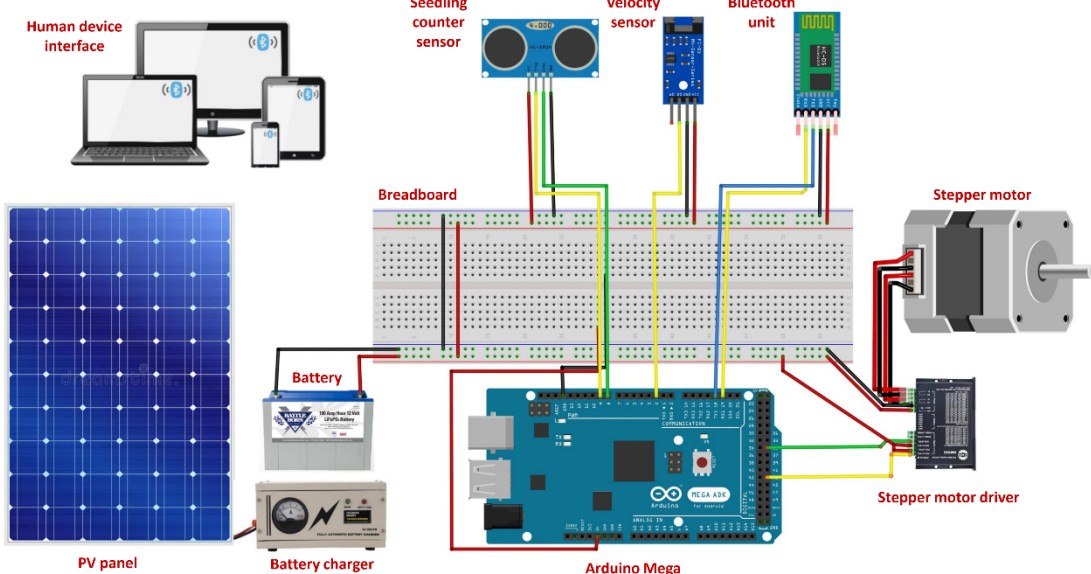

**Figure 3.** Schematic diagram of the VRCMM and the RSMS with successful electrical connections.

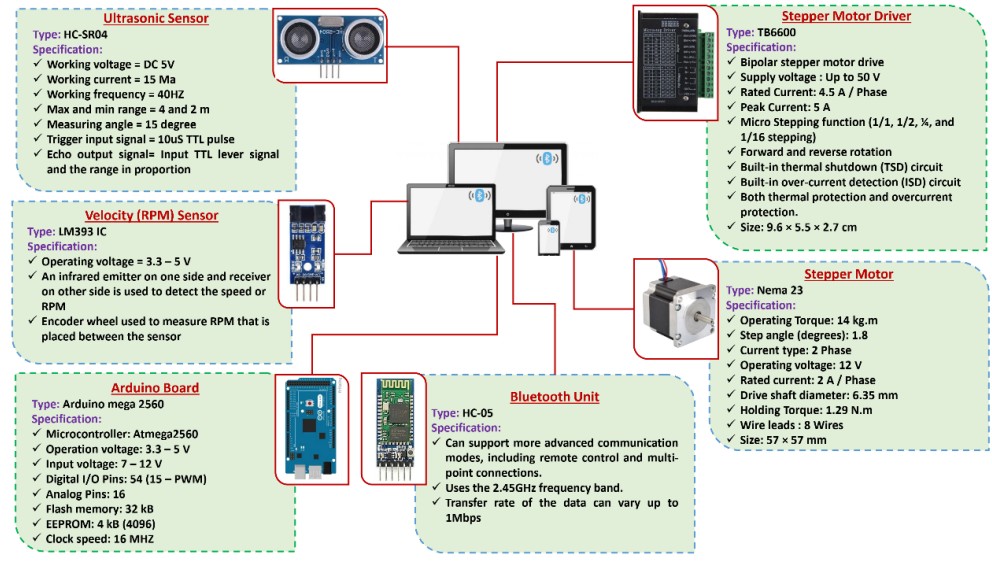

**Figure 4.** Specification of the main electronic components of the VRCMM and the RSMS.

### 2.3.2. Ultrasonic Sensor (Seedling Counter Sensor)

The ultrasonic sensor (model: HC-SR04) is a simple seedling counter, and it is also available in local markets. It consists of a crystal oscillator and both a transmitter and receiver for ultrasonic waves, as shown in Figure 6. When triggered, the transmitter releases eight bursts of a directed 40 kHz of ultrasonic waves and starts a timer.

The ultrasonic pulses move outward until they encounter the SS, as shown in Figure 7. The SS causes the ultrasonic pulses to be reflected directly towards the receiver. The ultrasonic receiver detects the reflected ultrasonic pulses and stops the timer. The ultrasonic burst speed is 340 m/s in air. The distance (*d*) between the SS and transmitter can be identified based on the number of counts made by the timer. The time (*T*), rate, and distance (TRD) measurement formulae are expressed in Equation (1).

$$d = C \times T \tag{1}$$

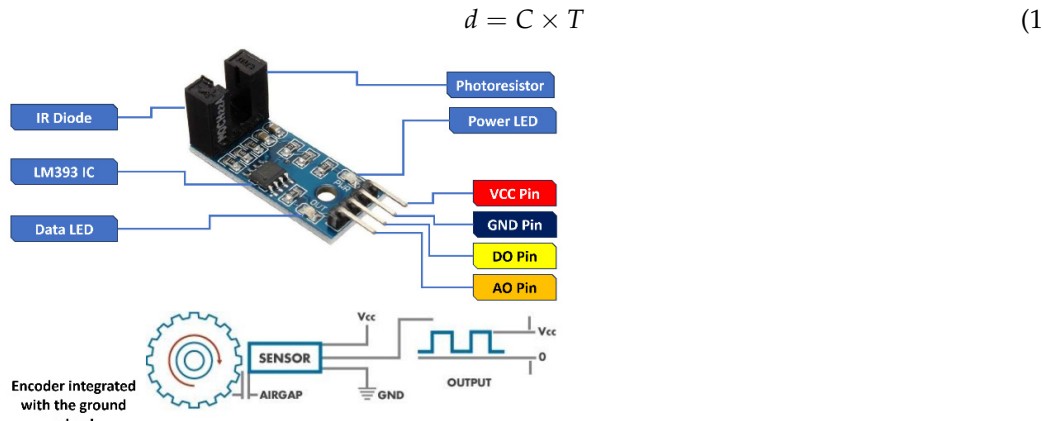

**Figure 5.** Main parts and connecting pins of the speed sensor used in the VRCMM. AO represents the analog pin, DO denotes the digital output pin, GND represents the ground pin, and VCC is the voltage common collector.

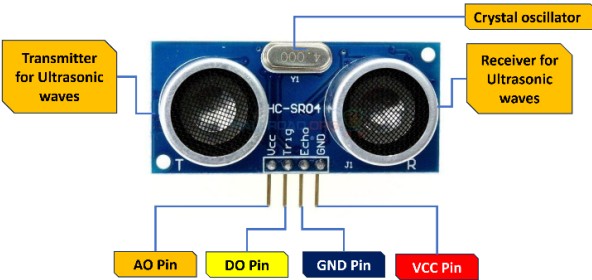

**Figure 6.** Main parts and connecting pins of the ultrasonic sensor used in the RSMS. AO represents the analog pin, DO denotes the digital output pin, GND represents the ground pin, and VCC is the voltage common collector.

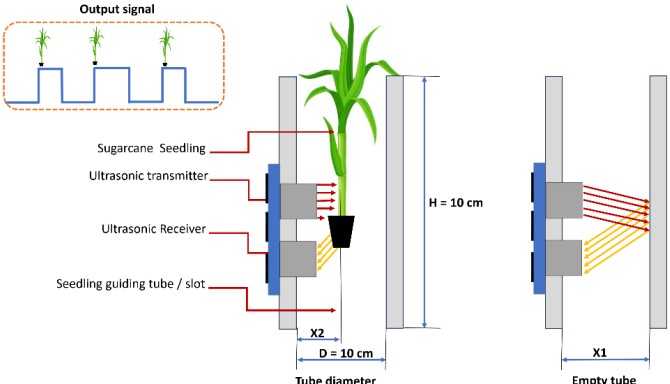

**Figure 7.** The main component and dimensions of the RSMS, where H is the height of SGS, D is the diameter of SGS, and X1 and X2 are the measured distances.

In this particular case, as it typically takes twice as long to record from transmitter to SS and then return to the receiver, the time is divided by two.

### 2.4. Variable-Rate Control Metering Mechanism (VRCMM)

The hardware components of the VRCMM were composed of four main electronic parts including an AMB, a speed sensor, stepper motor, and its driver, as shown in Figure 8.

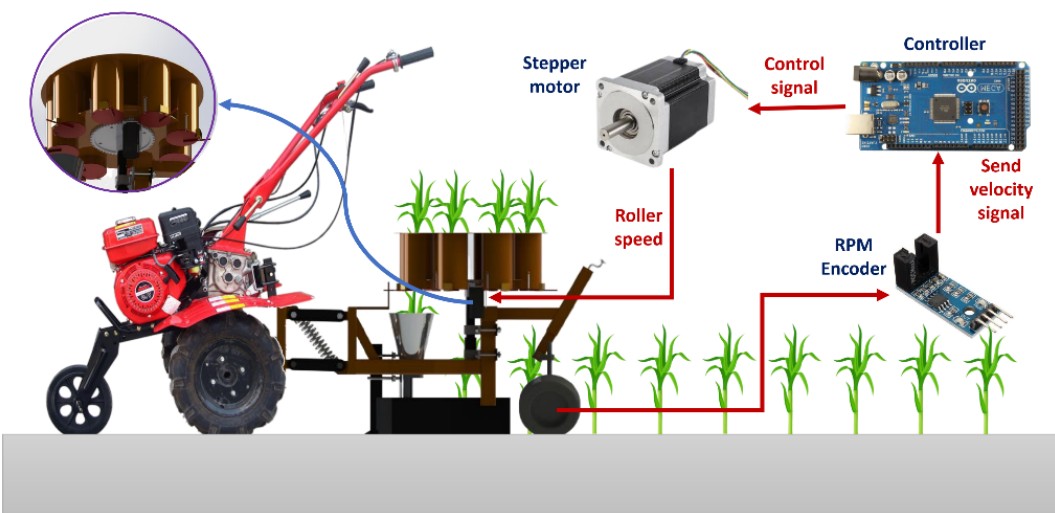

**Figure 8.** The schematic diagram of the VRCMM (Open-loop System).

2.4.1. Forward Speed

To design a precision sugarcane transplanting machine, the actual forward speed needs to be precisely measured. Where slippage occurred between the 2-wheel walking tractor and the soil, the speed sensor and encoder were connected to the ground wheel of the transplanter unit to identify its rotating and forward speed, as depicted in Figure 8. The ground wheel's RPM speed was detected, and the speed sensor supplied the output pulses to the AMB by a fixed time interval $\Delta T$, where the travel speed $v_f$ was calculated using Equation (2), according to Li et al. [49].

$$v_f = \frac{\pi \times d_2 \times C}{n \times \Delta T} \tag{2}$$

where $d_2$ is the ground wheel diameter in m, $C$ denotes the quantity of pulses in $\Delta T$, and $n$ is the encoder resolution in pulses per revolution.

2.4.2. Operating Algorithm of the VRCMM

The operating algorithm of the VRCMM with an open-loop system is shown in Figure 9. The AMB (micro-controller) is the essential part of the complete control system. The data (signal) obtained by the speed sensor, the transplanting operation parameters (i.e., seedling type, seedling characteristics, appropriate speed ratio, interval space, ground wheel diameter, and slip ratio) set by Arduino IED software (version 2.2.1), and the programming code were uploaded to the AMB before using a USB cable. The stepper motor driver transmits the driving instructions of the stepper motor after uploading the required information for each component. The operating algorithm in Figure 9 is employed to read, control, and analyze the transplanting process. The open-loop algorithm was employed to manage the stepper motor speed according to the travel and forward speed, and the control system saved and transferred the transplanting and travel and forward speed to a laptop or smartphone application via a Bluetooth unit.

*2.5. Seedling Counter System (SCS)*

The SCS consisted of an SGS and a seedling cylinder as part of the seedling metering device. The SGS was developed to allow the falling seedlings to pass steadily through the ultrasonic waves. In addition, the seedling rotating tube can help seedlings to fall through the SGS smoothly, as shown in Figure 10.

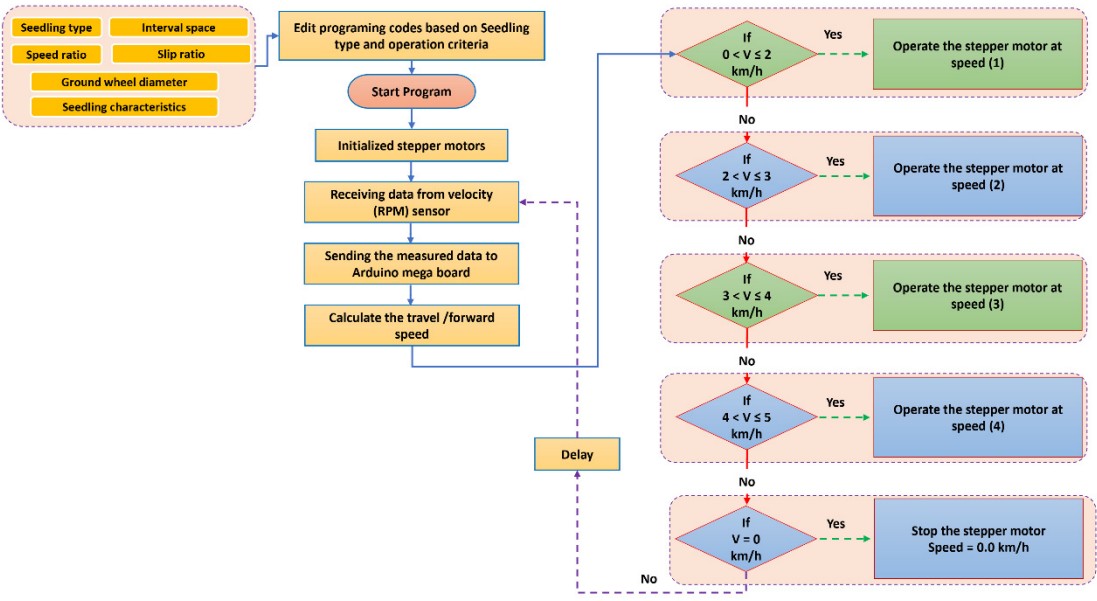

**Figure 9.** Operating algorithm flow chart of the VRCMM (open-loop system), where the first speed is 0–2 km/h, the second speed is 2–3 km/h, the third speed is 3–4 km/h, and the fourth speed is 4–5 km/h.

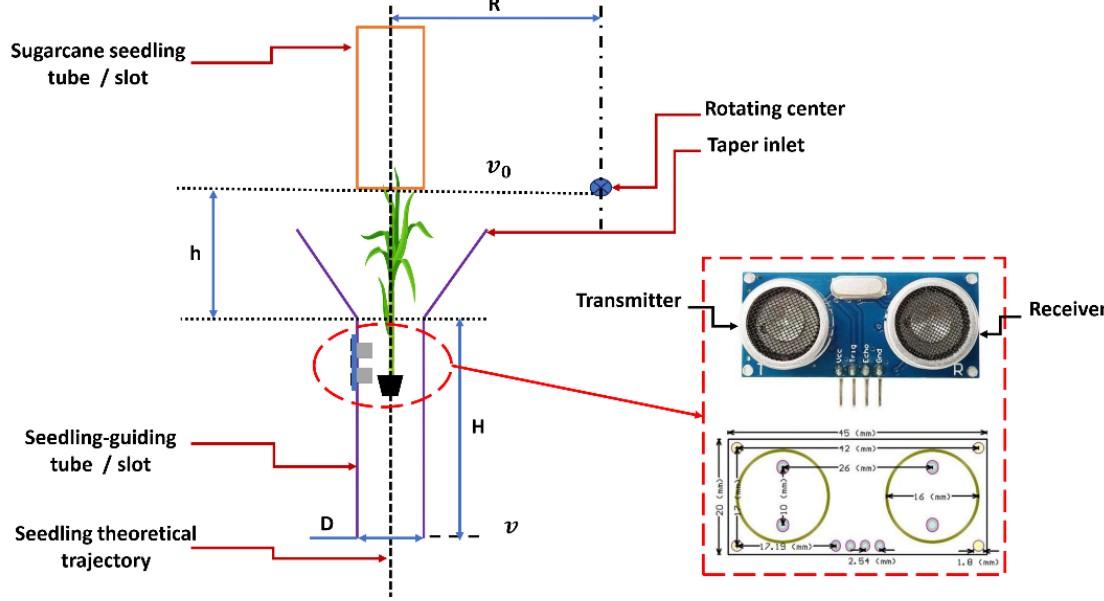

**Figure 10.** The falling trajectory when a seedling falls through the SGS, where R is the radius of the seedling cylinder, h is the vertical distance between the seedling cylinder and the taper inlet, H is the length of the SGS, D is the diameter of the SGS, $v_0$ is the initial velocity of the seedlings, and $v$ is the final velocity of the seedling.

### 2.5.1. Calculation of SCS

As seen in Figure 11, when the SS was released by the seedling cylinder, its projected path was a parabola curve. The vertical distance between the inlet and the outlet of the SGS was 35 mm.

The rotational speed of the seedling metering mechanism can be altered by adjusting the pulse-width modulation (PWM) of the stepper motor speed controller. The rotation speed of each seedling metering device matches to each combination of transplanting speed and spacing. The rotation speed of the seedling metering mechanism can be calculated using Equation (3), according to Xie et al. [50].

$$N_m = \frac{5000 \times v_f}{3 \times x_p \times n_c} \tag{3}$$

where $N_m$ refers to the rotation speed of the seedling metering device, r/min, $v_f$ refers to the forward speed of the transplanter, km/h, and the $x_p$ represents the theoretical transplanting distance, cm.

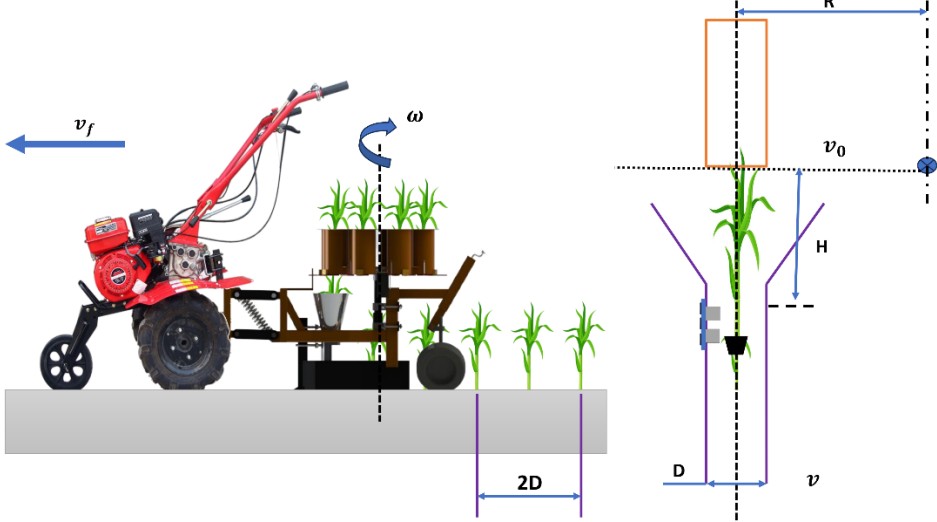

**Figure 11.** The falling trajectory when a seed falls through the SGS, where $v_0$ and $v$ are the initial and final velocities of the seedling, respectively, $v_f$ is the travel/forward speed, D is the transplant spacing, and $\omega$ is the rotating speed of the seedling metering device in rpm.

At the same time, the initial and final velocities of the SS can be calculated by using Equations (4)–(6) according to Xie et al. [50].

$$H = v_0 \times t + \frac{1}{2}g \times t^2 \tag{4}$$

where $H$ is the height of seedling-guiding slot, m, $v_0$ is the initial speed of the seedling, m/s, and $t$ is the time it takes for the seedling to reach the sensor from the seed outlet, s.

$$v = v_0 + g \times t \tag{5}$$

where $v_0 = 0$, $v$ is the speed of the seedling passing the sensor, m/s, and $g$ is the gravity acceleration = 9.81 m/s$^2$.

$$v = \sqrt{2g \times H} \tag{6}$$

### 2.5.2. Design of a Seedling Counter System (SCS)

The controller serves as a connection point between the seedling counter sensor and the software used for terminal display (on a laptop or smartphone), and it fundamentally performs the operating process as follows. Identifying and processing the electric signals required to have the target factors comes first. The next step is to achieve two-way real-time communication with the laptop or mobile phone and provide the transplanting factors to the terminal display software for real-time display, as well as the parameters that are returned by the monitoring terminal.

In this work, a laptop was utilized to develop a controller for the processor, which includes two main components: hardware parts and software program algorithms. The main job of the controller is illustrated in Figure 12. The process starts by acquiring the electrical signal created when the SS passes by the seedling counter sensor; the required parameters (number of seedlings, interval time between two consecutive seedlings, number of qualified seedlings, number of missed seedlings, qualified rate, and missed rate) are determined by the software program, which then assigns the parameter to the final variable

and sends it to the communication serial port according to the communication protocol. The Bluetooth communication module (HC-05) is connected the serial port, and the collected data are delivered there in real-time display by the laptop program. Figure 12 illustrates the operating algorithms for the seedling counter system; all the equations listed in the same figure were described by Tang at al. [51].

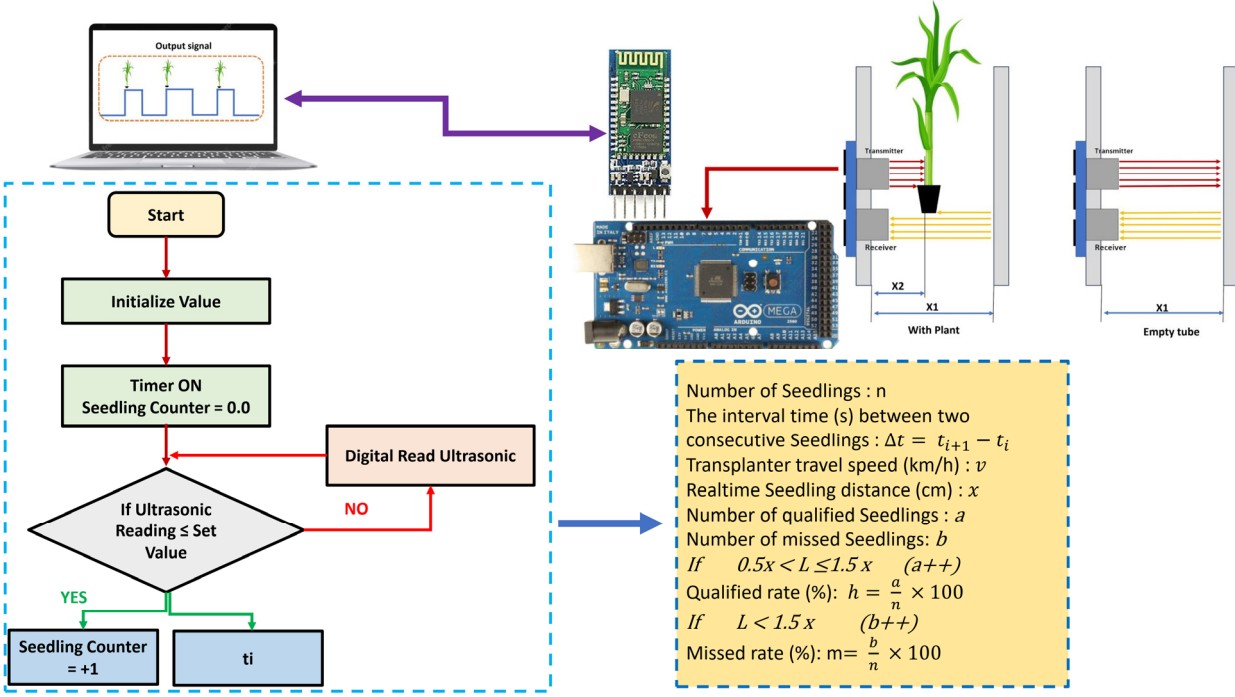

**Figure 12.** Operating algorithms for seedling flow rate detection, where *L* is the transplant spacing.

### 2.6. Remote Smart Monitoring System (RSMS)

In this investigation, the Arduino C programming languages, Mit-App inventor program, and Arduino IDE were used for the creation and design of the laptop application program. The laptop application can be used as seedling parameter monitoring software. It is capable of inputting the parameter values defined by the settings or the programming codes or display the monitored seedling factors in real time. In this instance, only one row of the transplanting parameter monitoring interface is employed because this is the seedling monitoring software for the single-row sugarcane transplanter. The software mainly comprises formulae like communication factor setting, transplanting parameter setting, and real-time transplanting factor display. The software's data update in real-time less frequently one in a second. Figure 13 shows the operating algorithm for the RSMS.

### 2.7. Laboratory Experiment

Comparative experiments were performed on a fabricated bench test to verify the effectiveness of the VRCMM and RSMS, at the Faculty of Agriculture and Natural Resources, University of Aswan, Egypt. In this manner, non-experimental variables such as slippage ratio and travel speed can properly be controlled, while the experimental levels can be altered efficiently. The Arduino Mega Controller was updated with the seedling information and programming codes. These data included the seedling type and characteristics, speed ratio, interval space, ground wheel diameter, slip ratio between the soil surfaces and the ground wheel, the theoretical seedling rate, the uploading time, the real seedling rate, and the regulating transplanting rate. Each experiment's operating time was determined by the time it took the virtual transplanter to complete a transplanting plot. The experimental data can be displayed on a laptop by a USB cable and saved in Excel files or displayed on the designed application only using serial communication software (Bluetooth). The examination bench was built of a seedling metering device, a stepper motor, a stepper

motor driver, a pinion and sprocket, a speed sensor, an AMB, and a seedling counter sensor. Furthermore, a taper inlet, an SGS, and metallic frames made up the seedling flow rate sensor.

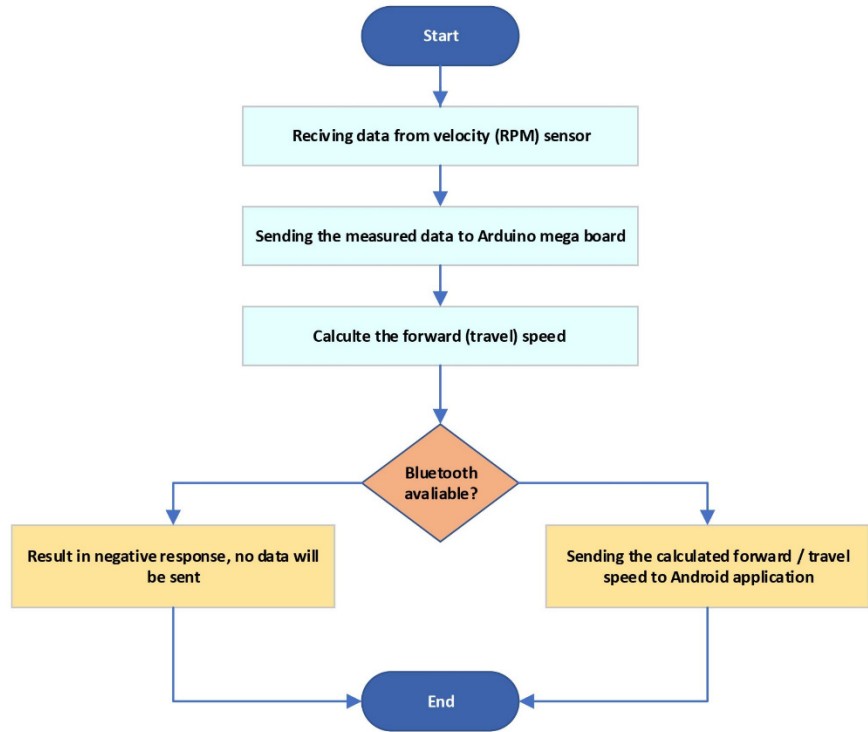

**Figure 13.** Operating algorithm for the RSMS.

### 2.8. Calibrating and Performance Evaluation of the VRCMM

The purpose of this test was to calibrate and assess the speed sensor's performance of the VRCMM under varying travel speeds and transplant spacing values. It aimed to find the rotating speed of the metering mechanism and the theoretical and actual transplant spacing under various seedling speeds. In addition, the test was run to verify the speed sensor accuracy of the seedling metering device at varying forward speeds and estimating relative error.

### 2.9. Performance Evaluation of the RSMS

This test examines the ultrasonic sensor's accuracy for monitoring seedling parameters at different travel speeds and transplant spacings. The goal was to find out the change law of the ultrasonic sensor's monitoring accuracy of single seedling flow at varying travel speeds and transplant spacings. It was run to discover the enhancement of precision of the ultrasonic sensor under the case of high travel speed and short transplant spacing. The test design is shown in Table 1. There was a total of twenty combinations (5 travel speeds × 4 transplant spacings), each of which was replicated three times, and each test had eight SSs.

**Table 1.** RSMS test design.

| Speed Level | 1 | 2 | 3 | 4 | 5 |
|:---:|:---:|:---:|:---:|:---:|:---:|
| Travel speed (km/h) | 2 | 3 | 4 | 5 | 6 |
| Transplant spacing (cm) | 20 | 25 | 30 | 40 | |

### 2.10. Statistical Analysis

A few statistical parameters were used, including coefficient of determination ($R^2$), to assess the performance of the speed sensor used in measuring processes of the travel

speed. Comparisons were performed among the data gathered from the speed sensor and the relevant (reference) data recorded by the stepper motor. $R^2$ denotes how much of the relationship between the measured and observed data is explained. Coefficient of determination ($R^2$) was computed using Equation (7), following the formula by Malone et al. [52].

$$R^2 = \frac{\sum(X - Y)^2}{\sum(X - Y_{ave})^2} \tag{7}$$

All parameters are explicated as follows: X is the independent value, Y is the dependent value, and $Y_{ave}$ is the average value.

The relative error ($E_R$) is the ratio of the absolute error to the target or desirable speed, where absolute error is the difference between actual or measured speed and the target or desirable speed. By employing this statistical technique, we may estimate the magnitude of the absolute error relative to of the real measurement size. It provides an indication clue of how accurately the measurement is related to the dimension of the object being measured, which can be computed using Equation (8).

$$E_R = \frac{E_A}{V_T} \times 100 = \frac{|V_a - V_T|}{V_T} \times 100 \tag{8}$$

where $E_A$ is the absolute error, $V_a$ is the target speed, $V_a$ is the actual or measured speed, km/h, and $V_T$ is the target or desirable speed, km/h.

The coefficient of variance ($CV$) was determined to assess the seedling accuracy and seedling consistency of different experimental techniques. The computational methods of the abovementioned indicators are described in Equation (9), according to Liu et al. [53]:

$$CV = \frac{SD}{S_{ave}} \times 100 \tag{9}$$

where $SD$ is the standard deviation and $S_{ave}$ is the mean value.

Using the statistical software IBM SPSS Statistics version 25 and Microsoft Office (Excel 365, Version 2016), one-way analysis of variance (ANOVA) was run on the collected data at a 0.05 significance level.

### 3. Results and Discussion

*3.1. Testing of the VRCMM*

For validation in a precision sugarcane transplanter, the newly developed electric-driven control metering mechanism and smart monitoring system were tested under laboratory conditions. The stepper motor driver was operated based on the received signals from the AMB, which are a function of the travel speed. The speed sensor's output signals have to be calibrated by using the standard instrument to ensure the quality of data acquisition.

Before deploying the speed sensor for the VRCMM, the sensor was calibrated against the real speed of the stepper motor integrated with the ground wheel. The calibrations of the sensors were performed for the speed sensor (model: HC-89 RPM), as illustrated in Figure 14. The travel speed is plotted on the x-axis, and for every calibration condition, the speed recorded by the speed sensor was plotted on the y-axis.

Figure 14 shows the validation of the speed sensor measurements. With high $R^2$ values of 0.9986, the measured speed by the speed sensor has been validated with the observed measures by the stepper motor attached to the ground wheel, demonstrating an ideal fit between the travel speed and the measured speed by the speed sensor. At various forward speeds, the speed sensor performed well, and it had strong linear regressions (y = 1.001x) that nearly overlapped the 1:1 line.

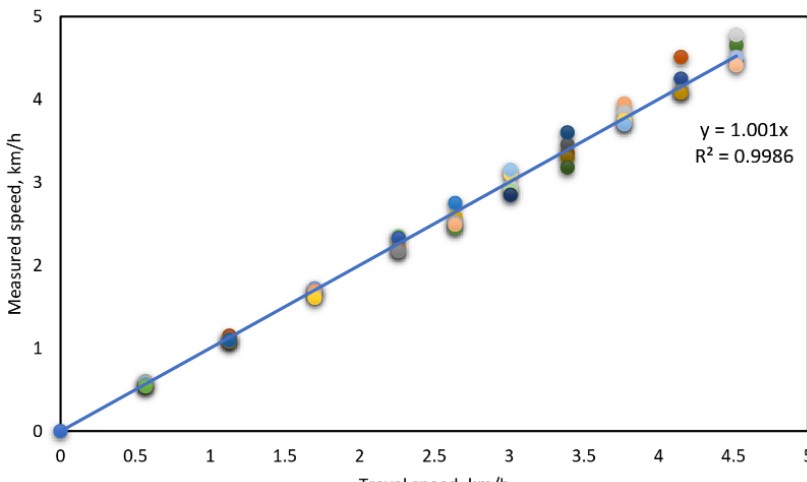

**Figure 14.** Tests on the speed regulation accuracy of the speed sensor of the VRCMM.

The accuracy of the stepper motor of the VRCMM's control system has a direct impact on the transplanting process. In the field, the transplanting process is controlled not only by the control system but also through the VRCMM's performance, the type of soil, soil state, slide between wheels and ground, etc. However, the bench test was mainly performed to detect the stability of the control system to keep the negative impact of the control system on the transplanting process at a minimum. As shown in Figure 15, the bench test was performed at 30 cm transplant spacing, and the travel speed of the ground wheel was selected between 0.5 km/h and 5 km/h, respectively. As depicted in Figure 16, the average relative error between the specified value of the VRCMM's stepper motor speed and the real value was 3.39%, and it increased with increasing the travel speed. The results further revealed that speed regulation was related with the transplanting index as reported by Che et al. [54], where they constructed a through-beam sensor with infrared sensing technology. The test findings revealed that the monitoring accuracy's relative error was between 0.5% and 5%.

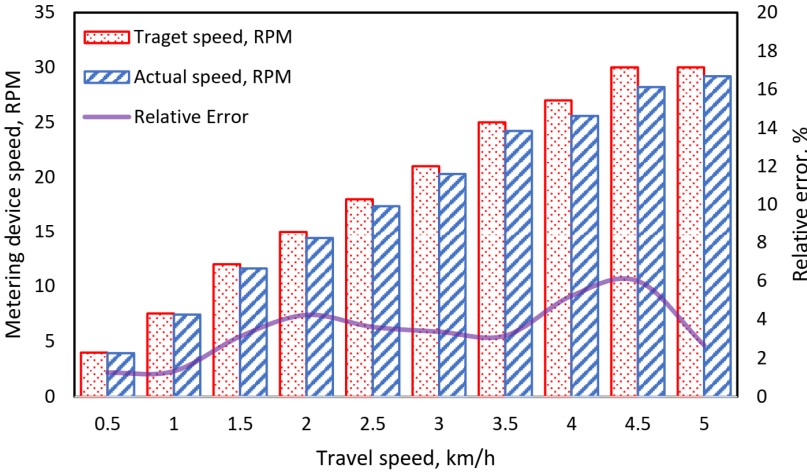

**Figure 15.** Speed accuracy detection of the VRCMM on bench test.

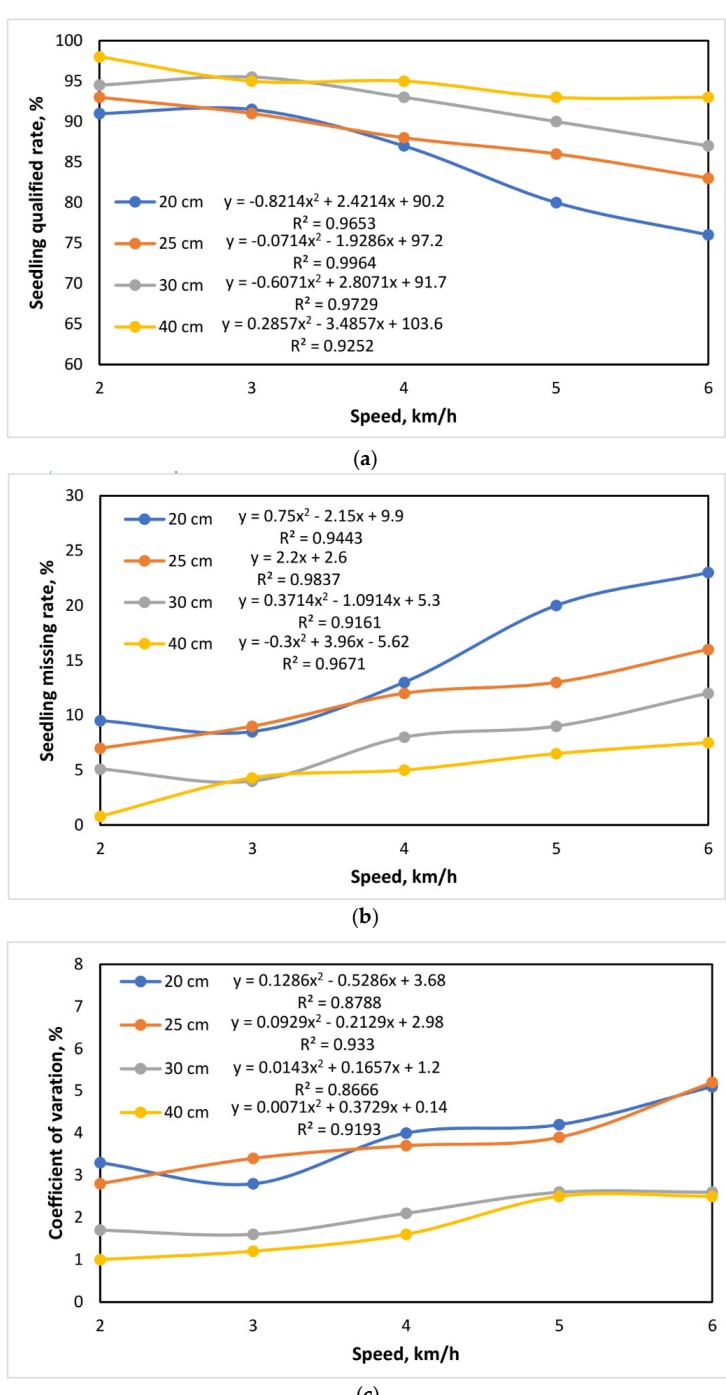

**Figure 16.** (**a**) Coefficient of determination of the quadratic relationships between travel speed on the RSMS and seedling qualified rate at different transplant spacings. (**b**) Coefficient of determin-tion of the quadratic relationships between travel speed on the RSMS and seedling missing rate at different transplant spacings. (**c**) Coefficient of determination of the quadratic relationships between travel speed on the RSMS and coefficient of variation at different transplant spacings.

### 3.2. The Effect of Travel Speed on the RSMS Results

Cay et al. [55] and Cay et al. [56] developed an electro-mechanic drive system (EMDS) for seed metering units. The findings of the field studies revealed that the speed of the seed plate is among the most crucial factors influencing the consistency of the planter's seed spacing. To estimate the impact of the travel speed on the obtained findings of the RSMS, the regression analysis of the travel speed was successfully performed under laboratory

conditions at different transplanting spacings (20, 30, 40, and 50 cm) between seedlings at the same row. According to the preliminary analysis of the bench test outcomes, the accuracy of seedling counter sensor monitoring is least affected by travel speed within the range of 2–4 km/h. The bench test findings are relatively consistent, and also travel speed has remarkably affected the accuracy, in the range of 4–6 km/h. The obtained data from the bench test showed that travel speed has a greater impact on RSMS accuracy. As a result, the essential record of 4 km/h is chosen to execute the segmented regression analysis. Figure 16 demonstrates the regression analysis of the seedling counter sensor on three evaluation parameters at different transplanting spacings. Furthermore, Figure 16a shows that the regression fitting grade of the travel speed varies with transplant spacing. Overall, it is possible to assume that the $R^2$ is the same in terms of order, reaching $R^2$ 20 cm > $R^2$ 25 cm > $R^2$ 30 cm > $R^2$ 35 cm. Furthermore, the regression fit is best when the transplant spacing is 25 cm. This proves that when the transplant spacing is short, the change in travel speed has a significant impact on the RSMS results. The RSMS accuracy of the seedling counter sensor decreases as the travel speed increases. This is explicable by the fact that, when the travel speed increases, so does the rotation speed of the seedling metering device of the VRCMM increases, as well as an increase in the number of seedlings released per time period. The seedling counter sensor's RSMS accuracy decreases as the travelling speed of the seedlings increases concurrently. Broadly, the RSMS accuracy of the seedling counter sensor is remarkably affected when the travel speed is high.

Xie et al. [44] reported that the remote tracking system's monitoring performance is greater at slower speeds, and its monitoring error increases as the speed increases, and the speed has an effect on its precision. According to Xie et al. [50], when the seeding speed becomes high, the sensor monitoring reliability changes dramatically. Wu et al. [57] created a portable seed measuring system based on a programmable controller (PLC) and a touchscreen. The test results show that the monitored seed metering system provides the greatest seeding effect when the planter's forward speed is set to 5.23 km/h. Yang et al. [58] found that as the speed increases, the qualified index decreases, and the missed seed index increases. According to Karimi et al. [59], a faster seeding speed will lower the sensor's accuracy to monitor process.

*3.3. The Effect of Transplant Spacing on the RSMS Results*

To assess the impact of transplant spacing on RSMS monitoring resultant outcomes, a transplant spacing regression analysis was performed at different travel speeds (2, 3, 4, 5, and 6 km/h). Similarly, preliminary examination of bench test results shows that in the 20–25 cm range, the bench test results change little, and transplant spacing has a bigger impact on RSMS accuracy; in the 25–40 cm range, the transplant spacing has a greater impact on RSMS accuracy. The bench test findings are reasonably steady, and transplant spacing has a significant impact on the seedling counter sensor's RSMS accuracy. As a result, the crucial point for segmented regression analysis was set at 25 cm transplant spacing. Figure 17 shows the regression analysis of the seedling counter sensor based on three criteria for assessment. Figure 17 makes it clear that the three assessment parameters of the seedling counter sensor have a similar law of change for transplant spacing, falling between 20 and 25 cm. Within the 25–40 cm range, there is a significant variance in the RSMS results; elsewhere, the changes are very consistent. This demonstrates that the RSMS accuracy is inconsistent and unsatisfactory in the 20–25 cm transplant spacing range. The RSMS accuracy is steady and good for transplant spacings between 25 and 40 cm. Furthermore, Figure 17 shows that the transplant spacing's degree of regression fitting varies with trip speeds. Overall, it can be concluded that the $R^2$ basically remains the same for all size orders: $R^2$ 2 km/h, $R^2$ 3 km/h, $R^2$ 4 km/h, $R^2$ 5 km/h, and $R^2$ 6 km/h. Additionally, the regression fit is the best when the travel speed is 2 km/h. This proves that the change in transplant spacing has a greater impact on the RSMS results when the travel speed is low. The RSMS accuracy of the seedling counter sensor decreased as the transplant spacing fell, which may be a result of the fixed number of cylinders in the seedling cylinder.

The speed of the seedling metering mechanism of the VRCMM can only be raised if the transplant spacing decreases to discharge more seedlings, and the number of seedlings discharged per unit time increases. The moving speed of the seedlings elevates in the meantime, which results in a decrease in the RSMS performance of the seedling counter sensor. In the case of small transplant spacing, the RSMS performance of the seedling counter sensor is impacted.

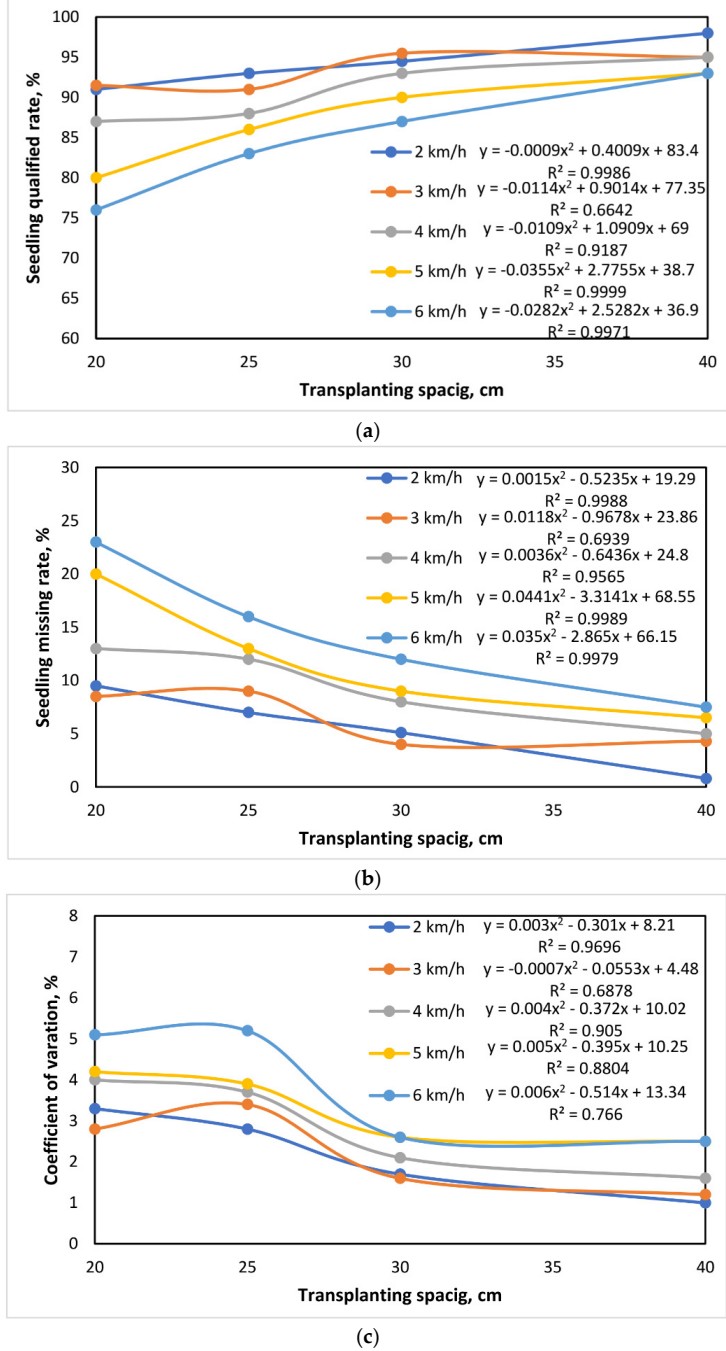

**Figure 17.** (**a**) Coefficient of determination of the quadratic relationships between transplant spacing on the RSMS and seedling qualified rate at different travel speeds. (**b**) Coefficient of determination of the quadratic relationships between transplant spacing on the RSM and seedling missing rate at different travel speeds. (**c**) Coefficient of determination of the quadratic relationships between transplant spacing on the RSM and coefficient of variation at different travel speeds.

Xie et al. [50] discovered that in cases where the seeding spacing is small, the sensor monitoring reliability changes dramatically. Further investigation revealed that the frequency of seeds going through the sensor is the most direct factor affecting sensor monitoring performance. The sensor monitoring reliability slowly decreases as frequency increases. Yang et al. [58] stated that speed has a significant impact on seeding performance, indicating that when speed increases, grain spacing dispersion becomes more extreme, the qualified index lowers, and the missing seed index rises.

## 4. Conclusions and Future Work

The potential of designing an automated sugarcane transplanter was investigated in this research. The study aimed to design and test the accuracy of a variable-rate control metering mechanism (VRCMM) and a remote smart monitoring system (RSMS) for a precision sugarcane transplanter based on IoT technology. The findings of the laboratory tests demonstrated the following:

- The speed sensor calibration results showed good performance, indicating perfect agreement between the travel speed and the speed measured by the speed sensor at different forward speeds during the experiment.
- The RSMS results are more affected by a change in travel speed when the transplant spacing is small. The RSMS accuracy of the seedling counter sensor decreased as the travel speed increased.
- When the travel speed is low, the change in transplant spacing has a greater impact on the RSMS outcomes. The RSMS accuracy of the seedling counter sensor decreased as the transplant spacing decreased.
- In conclusion, the obtained results of the current study are of great help to the design and improvement of a cost-effective, smart, and intelligent sugarcane transplanter.

**Author Contributions:** Conceptualization, A.E.E.; methodology, A.E.E., S.E., A.H.E. and A.H.M.; software, A.E.E., S.E., A.H.E., A.M.O. and Y.S.A.M.; validation, A.E.E., A.M.O., S.E., K.A.M., W.A.M., A.H.E. and Y.S.A.M.; formal analysis, A.E.E., A.M.O., A.H.M., K.A.M., W.A.M., S.E. and Y.S.A.M.; investigation, S.E., A.H.E. and A.H.M.; resources, A.E.E. and Y.S.A.M.; data curation, A.S.E., S.E., A.H.E., A.M.O., A.H.M., K.A.M., W.A.M. and Y.S.A.M.; writing—original draft preparation, A.E.E., S.E. and A.H.E.; writing—review and editing, A.E.E., Y.S.A.M., A.S.E., S.E., A.H.E., A.M.O., A.H.M., K.A.M., W.A.M. and visualization, A.E.E., S.E., A.H.E. and A.M.O.; supervision, A.H.E. and Y.S.A.M.; project administration, Y.S.A.M. and A.H.M.; funding acquisition, Y.S.A.M. All authors have read and agreed to the published version of the manuscript.

**Funding:** King Khalid University for funding this work through a large group Research Project under grant number (RGP 2/386/44).

**Institutional Review Board Statement:** Not applicable.

**Informed Consent Statement:** Not applicable.

**Data Availability Statement:** All data are presented within the article.

**Acknowledgments:** The authors would like to acknowledge the Deanship of Scientific Research at King Khalid University for funding this work through a large group Research Project under grant number (RGP 2/386/44).

**Conflicts of Interest:** The authors declare no conflict of interest.

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
