# Peer review of "Design and Validation of a Variable-Rate Control Metering Mechanism and Smart Monitoring System for a High-Precision Sugarcane Transplanter"

_agriculture, doi:10.3390/agriculture13122218_

Round 1

Reviewer 1 Report

Comments and Suggestions for Authors

The work entitled " Advanced Design and Performance Validation of a Variable-Rate Control Metering Mechanism and Smart Monitoring System for High Precision Sugarcane Transplanter Based on IoT Technology" was reviewed and comments and observations were made in order to improve its quality. This manuscript is really a very good prepared manuscript and contains serious consideration as potential publication. I think the work will contribute to the existing literature and should be published after a minor revision:

1.Keywords (Line 43): Avoid the words used in the title.

2. Results and discussion section: The results have been well represented but the discussion needs to be improved (i.e. comparison to other works and citations ).

3. It is advised to use the same font in the figures and text.

4. The multiple images in Figures 17 and 18 should be illustrated individually.

Author Response

Reviewer 1

We greatly appreciate your critical observations as well as your constructive and helpful comments. We hope that we could address your questions/comments by the explanations and revisions made in the manuscript. We believe that the manuscript is substantially improved after making the suggested revisions.

Comment 1:  Keywords (Line 43): Avoid the words used in the title.

Response: The authors are extremely thankful to the reviewer for this thoughtful point. As suggested by the esteemed reviewer, the keywords were adjusted. Kindly check the updated manuscript.                                               .

Comment 2: Results and discussion section: The results have been well represented but the discussion needs to be improved (i.e. comparison to other works and citations).

Response: The authors are extremely thankful to the reviewer for this thoughtful point. We improved this section by added the other previous studies. Kindly check the updated manuscript.                                               

Comments 3: It is advised to use the same font in the figures and text

Response: Many thanks for this comment. The font was adjusted. Kindly check the updated manuscript.                                               

Comments 4: The multiple images in Figures 17 and 18 should be illustrated individually.

Response: Many thanks for this comment. The mentioned Figures were illustrated individually. Kindly check the updated manuscript.                                                

The authors once again thank the learned Editors and Reviewers for their valuable comments for improving the quality of the manuscript.

Reviewer 2 Report

Comments and Suggestions for Authors

The article

Advanced Design and Performance Validation of a Variable Rate Control Metering Mechanism and Smart Monitoring System for High Precision Sugarcane Transplanter Based on IoT Technology

has been peer reviewed and the following considerations considered:

English is really bad.

For example: in line 49 eliminate sugary crops, particularly as well as in line 51 remove through elevating the sugarcane productivity.

In line 71 change to by for.

In line 87 it is crop rows.

The sentence following have to be rewritten (lines 87-90).

There is twice the word many in the same sentence in lines 94-95.

Line 96. Research is in singular or plural.

Photoelectric has to be with initial capital letter in line 114.

Change several and businesses by companies in line 123.

The letters in lines 133-139 seem of different size.

Line 144 add using a

What is FYM? The first time an acronym appears it has to be described completely.

In line 151 change a stepper motor driver by its driver.

In Figure 2, it has to be changed 12 to 5 converter.

Change in lines 181 and 187 rules by connections.

In line 191 eliminate is a simple and inexpensive way to

In line 199 change can work by works.

In line 205 eliminate and inexpensive way for measuring

In line 206 change counting by counter.

In line 209 change begins by starts.

The sentence in the heading of Fig. 7 should be rewritten as it is not clear (lines 223-225).

In line 228 change stepper motor driver by its driver.

Line 234 change needed by needs.

Which is the value of d2 and explain what it is in the text. What is n. All these constants appear in equation 2.

In line 247 it is missing by before using. Also a shoud be AN as it is before a vocal in USB.

Eliminate however in line 248.

Which are the 4 different speed in figure 9. Explain in text.

Eliminate rpm and add rotation speed in line 278. What is Nm, Vf, Nc and Xp in equation 3.

Is V the final speed given by equation 6?

Add the after the word between in line 291.

The end of the sentence after the word smartphone should be rewritten as it is not clear (lines 292-294).

In line 300 eliminate work and the semicolon.

Line 301 is not clear as there is no clarity in which parameter is assigned.

Eliminate “with is linked to the communication” and add to the in line 304-305.

Eliminate the C++ in line 312. Also eliminate as well as the

In line 313 eliminate applications served as the foundation for 313 the creation and design of the laptop application program.

THERE IS NO FIGURE 13.

Eliminate , whereas all laboratory tests were conducted in line 327.

In line 329 which are the non-experimental variables?

I cannot see in Table 1 the 20 different options and what you mean by level 1, 2, etc.

Add c in omparisons in line 361.

Line 385-388 have to be rewritten to clarify the idea.

Rewrite line 402 and 403.

What do you mean in the red square of Figure 16: target speed?

What you mean in line 422 by: The bench test findings are highly

Figure 17 is not clear. What is the meaning of 20, 30 and 40 cm? Is it the depth or the distance between transplanting? How it was detected as a miss?

This should be explained in the text.

There is a lot of text missing on the discussion section when it has to be compared with other transplants systems. It only gives the results but does not discuss the results when compared with another machine.

In the conclusion the hypothesis should not be included so remove lines 477-480.

Lines 481 to 489 are more results and discussion.

Only lines 490 to 495 are useful in the conclusion section.

Comments on the Quality of English Language

I add many comments of english details in the pdf file

Author Response

Reviewer 2

We greatly appreciate your critical observations as well as your constructive and helpful comments. We hope that we could address your questions/comments by the explanations and revisions made in the manuscript. We believe that the manuscript is substantially improved after making the suggested revisions.

Comment 1: English is really bad.

Response: The authors are extremely thankful to the reviewer for this thoughtful point. As suggested by the esteemed reviewer, English language was improved.  Kindly check the updated manuscript.

Comment 2: In line 49 eliminate sugary crops, particularly as well as in line 51 remove through elevating the sugarcane productivity.

Response: Many thanks for this comment. As suggested by the esteemed reviewer, the suggested comment was done.  Kindly check the updated manuscript in lines 55 & 57.

Comment 3: In line 71 change to by for.

Response: Many thanks for this comment. As suggested by the esteemed reviewer, the suggested comment was done. Kindly check the updated manuscript in line 73.

Comment 4: In line 87 it is crop rows.

Response: Many thanks for this comment. As suggested by the esteemed reviewer, the suggested comment was done.  Kindly check the updated manuscript in line 89.

Comment 5: The sentence following have to be rewritten (lines 87-90).

Response: Many thanks for this comment. As suggested by the esteemed reviewer, the suggested comment was done. Kindly check the updated manuscript in lines 89 to 93.

Comment 6: There is twice the word many in the same sentence in lines 94-95.

Response: Many thanks for this comment. As suggested by the esteemed reviewer, the suggested comment was done. Kindly check the updated manuscript in lines 100-101.

Comment 7: Line 96. Research is in singular or plural.

Response: Many thanks for this comment. As suggested by the esteemed reviewer, the suggested comment was done. Kindly check the updated manuscript in line 102.

Comment 8: Photoelectric has to be with initial capital letter in line 114.

Response: Many thanks for this comment. As suggested by the esteemed reviewer, the suggested comment was done. Kindly check the updated manuscript in line 117.

Comment 9: Change several and businesses by companies in line 123.

Response: Many thanks for this comment. As suggested by the esteemed reviewer, the suggested comment was done. Kindly check the updated manuscript in line 114.

Comment 10: The letters in lines 133-139 seem of different size.

Response: Many thanks for this comment. As suggested by the esteemed reviewer, the suggested comment was done. Kindly check the updated manuscript from line134 to line 139.

Comment 11: Line 144 add using a

Response: The authors are extremely thankful to the reviewer for this thoughtful point. As suggested by the esteemed reviewer, the suggested comment was done. Kindly check the updated manuscript in line 144.

Comment 12: What is FYM? The first time an acronym appears it has to be described completely.

Response: Many thanks for this comment. As suggested by the esteemed reviewer, the suggested comment was done. Kindly check the updated manuscript in line 146.

Comment 13: In line 151 change a stepper motor driver by its driver.

Response: Many thanks for this comment. As suggested by the esteemed reviewer, the suggested comment was done. Kindly check the updated manuscript in line 154.

Comment 14: In Figure 2, it has to be changed 12 to 5 converter.

Response: The authors are extremely thankful to the reviewer for this thoughtful point. As suggested by the esteemed reviewer, the suggested comment was done.

Comment 15: Change in lines 181 and 187 rules by connections.

Response: Many thanks for this comment. As suggested by the esteemed reviewer, the suggested comment was done. Kindly check the updated manuscript in line 182 and line 189.

Comment 16: In line 191 eliminate is a simple and inexpensive way to

Response: Many thanks for this comment. As suggested by the esteemed reviewer, the suggested comment was done. Kindly check the updated manuscript in line 193.

Comment 17: In line 199 change can work by works.

Response: Many thanks for this comment. As suggested by the esteemed reviewer, the suggested comment was done. Kindly check the updated manuscript in line 201.

Comment 18: In line 205 eliminate and inexpensive way for measuring

Response: Many thanks for this comment. As suggested by the esteemed reviewer, the suggested comment was done. Kindly check the updated manuscript in line 207.

Comment 19: In line 206 change counting by counter.

Response: Many thanks for this comment. As suggested by the esteemed reviewer, the suggested comment was done. Kindly check the updated manuscript in line 208.

Comment 20: In line 209 change begins by starts.

Response-20: Many thanks for this comment. As suggested by the esteemed reviewer, the suggested comment was done. Kindly check the updated manuscript in line 211.

Comment 21: The sentence in the heading of Fig. 7 should be rewritten as it is not clear (lines 223-225).

Response: Many thanks for this comment. As suggested by the esteemed reviewer, the suggested comment was done. Kindly check the updated manuscript in line 224.

Comment 22: In line 228 change stepper motor driver by its driver.

Response: Many thanks for this comment. As suggested by the esteemed reviewer, the suggested comment was done. Kindly check the updated manuscript in line 231.

Comment 23: Line 234 change needed by needs.

Response: Many thanks for this comment. As suggested by the esteemed reviewer, the suggested comment was done. Kindly check the updated manuscript in line 237.

Comment 24: Which is the value of d2 and explain what it is in the text. What is n. All these constants appear in equation 2.

Response: Many thanks for this comment. As suggested by the esteemed reviewer, the suggested comment was done. Kindly check the updated manuscript in line 244 & 245.

Comment 25: In line 247 it is missing by before using. Also a shoud be AN as it is before a vocal in USB.

Response: Many thanks for this comment. As suggested by the esteemed reviewer, the suggested comment was done. Kindly check the updated manuscript in line 252.

Comment 26: Eliminate however in line 248.

Response: Many thanks for this comment. As suggested by the esteemed reviewer, the suggested comment was done. Kindly check the updated manuscript in line 252.

Comment 27: Which are the 4 different speeds in figure 9? Explain in text.

Response: Many thanks for this comment. As suggested by the esteemed reviewer, the text was explained. Kindly check the updated manuscript in lines 259 to 261.

Comment 28: Eliminate rpm and add rotation speed in line 278. What is Nm, Vf, Nc and Xp in equation 3.

Response: Many thanks for this comment. As suggested by the esteemed reviewer, the suggested comment was done. Kindly check the updated manuscript in lines 280 & 282 and from lines 284 to 287.

Comment 29: Is V the final speed given by equation 6?

Response: The authors are extremely thankful to the reviewer for this thoughtful point. As suggested by the esteemed reviewer, the variable was defined under equation 6. Kindly check the updated manuscript in line 296.

Comment 30: Add the after the word between in line 291.

Response: Many thanks for this comment. As suggested by the esteemed reviewer, the suggested comment was done. Kindly check the updated manuscript in line 299.

Comment 31: The end of the sentence after the word smartphone should be rewritten as it is not clear (lines 292-294).

Response: Many thanks for this comment. As suggested by the esteemed reviewer, the suggested comment was done. Kindly check the updated manuscript in line302 to line 304.

Comment 32: In line 300 eliminate work and the semicolon.

Response: The authors are extremely thankful to the reviewer for this thoughtful point. As suggested by the esteemed reviewer, the suggested comment was done. Kindly check the updated manuscript in line 306.

Comment 33: Line 301 is not clear as there is no clarity in which parameter is assigned.

Response: Many thanks for this comment. As suggested by the esteemed reviewer, the parameters are mentioned. Kindly check the updated manuscript in line 308 & 309.

Comment 34: Eliminate “with is linked to the communication” and add to the in line 304-305.

Response: Many thanks for this comment. As suggested by the esteemed reviewer, the suggested comment was done. Kindly check the updated manuscript in line 312.

Comment 35: Eliminate the C++ in line 312. Also eliminate as well as the

Response: Many thanks for this comment. As suggested by the esteemed reviewer, the suggested comment was done. Kindly check the updated manuscript in line 320.

Comment 36: In line 313 eliminate applications served as the foundation for 313 the creation and design of the laptop application program.

Response: The authors are extremely thankful to the reviewer for this thoughtful point. As suggested by the esteemed reviewer, the suggested comment was done. Kindly check the updated manuscript in line 321.

Comment 37: THERE IS NO FIGURE 13.

Response: Many thanks for this comment. As suggested by the esteemed reviewer, the Figure number was modified in the text.

Comment 38: Eliminate, whereas all laboratory tests were conducted in line 327.

Response 38: Many thanks for this comment. As suggested by the esteemed reviewer, the suggested comment was done. Kindly check the updated manuscript in line 336.

Comment 39: In line 329 which are the non-experimental variables?

Response: The authors are extremely thankful to the reviewer for this thoughtful point. As suggested by the esteemed reviewer, the non-experimental variables were mentioned. Kindly check the updated manuscript in line 338.

Comment 40: I cannot see in Table 1 the 20 different options and what you mean by level 1, 2, etc.

Response: The authors are extremely thankful to the reviewer for this thoughtful point. As suggested by the esteemed reviewer, the suggested comment was done. Kindly check the updated manuscript in line 365.

Comment 41: Add c in omparisons in line 361.

Response: The authors are extremely thankful to the reviewer for this thoughtful point. As suggested by the esteemed reviewer, the suggested comment was done. Kindly check the updated manuscript in line 374.

Comment 42: Line 385-388 have to be rewritten to clarify the idea.

Response: The authors are extremely thankful to the reviewer for this thoughtful point. As suggested by the esteemed reviewer, the suggested comment was done. Kindly check the updated manuscript in lines 397 to 400.

Comment 43: Rewrite line 402 and 403.

Response: The authors are extremely thankful to the reviewer for this thoughtful point. As suggested by the esteemed reviewer, the suggested comment was done. Kindly check the updated manuscript in lines 417 to 418.

Comment 44: What do you mean in the red square of Figure 16: target speed?

Response: The authors are extremely thankful to the reviewer for this thoughtful point. As suggested by the esteemed reviewer, target speed is means the desirable or required speed. Kindly check the updated manuscript.

Comment 45: What you mean in line 422 by: The bench test findings are highly

Response: The authors are extremely thankful to the reviewer for this thoughtful point. As suggested by the esteemed reviewer, the science was re written.  Kindly check the updated manuscript in lines 447 to 448.

Comment 46: Figure 17 is not clear. What is the meaning of 20, 30 and 40 cm? Is it the depth or the distance between transplanting? How it was detected as a miss?

This should be explained in the text

Response: The authors are extremely thankful to the reviewer for this thoughtful point. As suggested by the esteemed reviewer, it is means the distance between transplanting and it was explained in the text. Kindly check the updated manuscript in lines 442 to 443.

Comment 47: There is a lot of text missing on the discussion section when it has to be compared with other transplants systems. It only gives the results but does not discuss the results when compared with another machine.

Response: The authors are extremely thankful to the reviewer for this thoughtful point. As suggested by the esteemed reviewer, the discussion section was improves.   Kindly check the updated discussion.

Comment 48: In the conclusion the hypothesis should not be included so remove lines 477-480.

Response: The authors are extremely thankful to the reviewer for this thoughtful point. As suggested by the esteemed reviewer, the hypothesis was removed. Kindly check the updated manuscript.

Comment 49: Lines 481 to 489 are more results and discussion.

Response: The authors are extremely thankful to the reviewer for this thoughtful point. As suggested by the esteemed reviewer, this part was removed.

Comment 50: Only lines 490 to 495 are useful in the conclusion section.

Response: The authors are extremely thankful to the reviewer for this thoughtful point.  The conclusion section was modified.

The authors once again thank the learned Editors and Reviewers for their valuable comments for improving the quality of the manuscript.

Reviewer 3 Report

Comments and Suggestions for Authors

The authors propose to “design and test the accuracy of a variable-rate control metering mechanism (VRCMM) and a remote smart monitoring system (RSMS) for a precision sugarcane transplanter based on IoT technology”. Without considering the description and justification of the VRCMM and RSMS design, the reviewer has questions about conducting the experiment, analysing and processing the obtained results:

1.    The test design, as shown in Table 1, originally specified five levels of travel speed (2 km/h, 3 km/h, 4 km/h, 5 km/h and 6 km/h). However, Figures 15 and 16 show a wider range of ten travel speed levels, ranging from 0.5 km/h to 5 km/h with 0.5 km/h step. What is the number of travel speed levels in the test design?

2.    The authors introduce the relative error, calculated as the absolute error of a measurement divided by the actual measurements, using the formula (8), and concluded, that the average relative error was 3.39%. The method for calculating the average relative error remains unclear (as seen in Figure 16), and it is not clear why the authors concluded that, that “the accuracy of speed regulation was in line with the transplanting index”.

3.    Figures 17 demonstrates the piecewise (segmented) regression analysis results for travel speed (km/h) four segments depending on four type of transplant spacing (cm). In turn Figure 18 demonstrates the piecewise regression analysis results for transplant spacing (cm) three segments depending on five type of travel speed (km/h). It is unclear why the authors opted for linear regression when the figures display nonlinear curves rather than straight lines?

4.    How the coefficient of determination for all data of segmented regression analysis was calculated?

5.    In each segment of the piecewise regression analysis, what is the number of data points available?

6.    The specific objectives of the study are identical to the main goal of the study and lack specific properties (line 133 – 139).

7.    The statement to “testify the efficiency of the three basic parameters of seedling amount, optimum rate, and missed rate”, as well as “the speed regulation efficiency” is not confirmed by the definition and justification of the concept of efficiency.

Furthermore, there are several technical comments:

1.    The title of the article is too long (25 words, including one abbreviation)

2.    Several sentences begin with a lowercase letter, or there is no period at the end of those sentence.

3.    All formulas and figures lack explanations for the included variables or symbols.

4.    The use of the coefficient of determination, denoted as R2, creates confusion for the reader when comparing it in lines 428 and 459. For instance, in line 428, R220cm>R225cm>R230cm>R235cm.

5.    The citation of source [52] with respect to the formula for computing the Pearson correlation is inaccurate.

Author Response

Reviewer 3

We greatly appreciate your critical observations as well as your constructive and helpful comments. We hope that we could address your questions/comments by the explanations and revisions made in the manuscript. We believe that the manuscript is substantially improved after making the suggested revisions.

Comment-1: The test design, as shown in Table 1, originally specified five levels of travel speed (2 km/h, 3 km/h, 4 km/h, 5 km/h and 6 km/h). However, Figures 15 and 16 show a wider range of ten travel speed levels, ranging from 0.5 km/h to 5 km/h with 0.5 km/h step. What is the number of travel speed levels in the test design?

Response: The authors are thankful to the honorable reviewer for the words of encouragement and trust in our work.  

Based on The objectives of the study, there are two different aims:

The first objective: design and assess the performance of a variable-rate control metering mechanism for sugarcane transplanting based on IoT technology. Which, it was evaluated at many different speeds (0.5, 1.0, 1.5, 2.0, 2.5, 3.0, 3.5, 4.0, 4.5, and 5.0 km/h) for evaluating the performance of the developed variable-rate control metering mechanism.

The second objective: Design and test a remote smart monitoring system (RSMS) for precise sugarcane transplanter based on ultrasonic sensor, IoT technology, Android, and wireless communication. Which, it was evaluated at only five travel speeds (2.0, 3.0, 4.0, 5.0, and 6.0 km/h) for evaluating the performance of the developed remote smart monitoring system.

Comment-2: The authors introduce the relative error, calculated as the absolute error of a measurement divided by the actual measurements, using the formula (8), and concluded, that the average relative error was 3.39%. The method for calculating the average relative error remains unclear (as seen in Figure 16).

Response-2: The authors are thankful to the honorable reviewer for the words of encouragement and trust in our work. The method for calculating the average relative error was rewritten. Kindly check the updated manuscript in page [13]

It is not clear why the authors concluded that “the accuracy of speed regulation was in line with the transplanting index”.

Many thanks for this comment. This sentence was modified since the relationships are not linear but it is quadratic relationships Kindly check the updated manuscript in page [15] lines 453 - 454.                   

Comment 3:  Figures 17 demonstrates the piecewise (segmented) regression analysis results for travel speed (km/h) four segments depending on four type of transplant spacing (cm). In turn Figure 18 demonstrates the piecewise regression analysis results for transplant spacing (cm) three segments depending on five type of travel speed (km/h). It is unclear why the authors opted for linear regression when the figures display nonlinear curves rather than straight lines?

Response-3: The authors are thankful to the honorable reviewer for the words of encouragement and trust in our work. We agree with you, we changed the regression analysis equation and recalculated the R2 with quadratic (non linear). Kindly check the updated manuscript (figure 16 & figure 17).               

Comment 4: How the coefficient of determination for all data of segmented regression analysis was calculated?

Response-4: The authors are thankful to the honorable reviewer for the words of encouragement and trust in our work. We changed the regression analysis equation. Kindly check the updated manuscript equation (7).

Comment-5: In each segment of the piecewise regression analysis, what is the number of data points available?

Response-5: The authors are thankful to the honorable reviewer for the words of encouragement and trust in our work.  (4 transplanting distances, 5 travel speeds, 3 replicates for each one). Kindly check the updated manuscript from lines 380 to 382 in page [13]               

Comment-6:  The specific objectives of the study are identical to the main goal of the study and lack specific properties (line 133 – 139).

Response-6: The authors are thankful to the honorable reviewer for the words of encouragement and trust in our work. This part was modified. Kindly check the updated manuscript from lines137 to 142 in page [3] .               

Comment-7: The statement to “testify the efficiency of the three basic parameters of seedling amount, optimum rate, and missed rate”, as well as “the speed regulation efficiency” is not confirmed by the definition and justification of the concept of efficiency.

Response-7: The authors are thankful to the honorable reviewer for the words of encouragement and trust in our work. We agree with you, so we changed it. Kindly check the updated manuscript.               

Furthermore, there are several technical comments:                                                                                                    

Comment-1: The title of the article is too long (25 words, including one abbreviation)

Response-1: The authors are thankful to the honorable reviewer for the words of encouragement and trust in our work. The title was adjusted. Kindly check the updated manuscript.

Comment-2: Several sentences begin with a lowercase letter, or there is no period at the end of those sentence.

Response-2: The authors are thankful to the honorable reviewer for the words of encouragement and trust in our work. The sentences were corrected. Kindly check the updated manuscript.               

Comment-3: All formulas and figures lack explanations for the included variables or symbols.

Response-3: The authors are thankful to the honorable reviewer for the words of encouragement and trust in our work.  All variables and symbols were explained. Kindly check the updated manuscript.               

Comment-4: The use of the coefficient of determination, denoted as R2, creates confusion for the reader when comparing it in lines 428 and 459. For instance, in line 428, R220cm>R225cm>R230cm>R235cm.

Response-4: The authors are thankful to the honorable reviewer for the words of encouragement and trust in our work.  The R2 values were adjusted. Kindly check the updated manuscript in pages [15 and 18].               

Comment-5: The citation of source [52] with respect to the formula for computing the Pearson correlation is inaccurate.

Response-5: The authors are thankful to the honorable reviewer for the words of encouragement and trust in our work.  We changed it by another source. Kindly check the updated manuscript in page [13].              

The authors once again thank the learned Editors and Reviewers for their valuable comments for improving the quality of the manuscript.

Round 2

Reviewer 3 Report

Comments and Suggestions for Authors

Even though the regression analysis equation has changed to a nonlinear model, the titles of figures 16 and 17 still state "...linear regression...," and the specific regression model equation is not provided. Additionally, could you clarify whether formulas 4, 5, and 6 include a period or a multiplication sign?

Author Response

Reviewer 3

We greatly appreciate your critical observations as well as your constructive and helpful comments. We hope that we could address your questions/comments by the explanations and revisions made in the manuscript. We believe that the manuscript is substantially improved after making the suggested revisions.

Comment: Even though the regression analysis equation has changed to a nonlinear model, the titles of figures 16 and 17 still state "...linear regression...," and the specific regression model equation is not provided.

Response: The authors are thankful to the honorable reviewer for the words of encouragement and trust in our work. Linear regression from the figures 16a, b & c as well as figures 17, b & c and titles were removed. The specific regression model equation was provided inside each figure (Pages 16, 17, 18 & 19).

Comment: Additionally, could you clarify whether formulas 4, 5, and 6 include a period or a multiplication sign?

Response: The authors are thankful to the honorable reviewer for the words of encouragement and trust in our work.  Formulas 4, 5, and 6 include multiplication sign. We have modified them at lines 288, 291, and 295 in page 10.

The authors once again thank the learned Editors and Reviewers for their valuable comments for improving the quality of the manuscript.
